# CD44 Modulates Cell Migration and Invasion in Ewing Sarcoma Cells

**DOI:** 10.3390/ijms241411774

**Published:** 2023-07-21

**Authors:** Enrique Fernández-Tabanera, Laura García-García, Carlos Rodríguez-Martín, Saint T. Cervera, Laura González-González, Cristina Robledo, Santiago Josa, Selene Martínez, Luis Chapado, Sara Monzón, Raquel M. Melero-Fernández de Mera, Javier Alonso

**Affiliations:** 1Unidad de Tumores Sólidos Infantiles, Instituto de Investigación de Enfermedades Raras (IIER), Instituto de Salud Carlos III (ISCIII), 28220 Madrid, Spain; efernandezt@isciii.es (E.F.-T.);; 2Centro de Investigación Biomédica en Red de Enfermedades Raras, Instituto de Salud Carlos III (U758, CB06/07/1009, CIBERER-ISCIII), 28029 Madrid, Spain; 3Universidad Nacional de Educación a Distancia (UNED), 28015 Madrid, Spain; 4Bioinformatics Unit, Instituto de Salud Carlos III (ISCIII), 28220 Madrid, Spain

**Keywords:** Ewing sarcoma, CD44, EWSR1::FLI1, cell migration, cell invasiveness

## Abstract

The chimeric EWSR1::FLI1 transcription factor is the main oncogenic event in Ewing sarcoma. Recently, it has been proposed that EWSR1::FLI1 levels can fluctuate in Ewing sarcoma cells, giving rise to two cell populations. EWSR1::FLI1^low^ cells present a migratory and invasive phenotype, while EWSR1::FLI1^high^ cells are more proliferative. In this work, we described how the CD44 standard isoform (CD44s), a transmembrane protein involved in cell adhesion and migration, is overexpressed in the EWSR1::FLI1^low^ phenotype. The functional characterization of CD44s (proliferation, clonogenicity, migration, and invasion ability) was performed in three doxycycline-inducible Ewing sarcoma cell models (A673, MHH-ES1, and CADO-ES1). As a result, CD44s expression reduced cell proliferation in all the cell lines tested without affecting clonogenicity. Additionally, CD44s increased cell migration in A673 and MHH-ES1, without effects in CADO-ES1. As hyaluronan is the main ligand of CD44s, its effect on migration ability was also assessed, showing that high molecular weight hyaluronic acid (HMW-HA) blocked cell migration while low molecular weight hyaluronic acid (LMW-HA) increased it. Invasion ability was correlated with CD44 expression in A673 and MHH-ES1 cell lines. CD44s, upregulated upon EWSR1::FLI1 knockdown, regulates cell migration and invasion in Ewing sarcoma cells.

## 1. Introduction

Ewing sarcoma is a rare and aggressive tumor that arises mainly during childhood and adolescence, predominantly in the bones of the proximal and distal extremities [1]. Approximately 25% of patients have detectable metastases at the time of diagnosis. Molecularly, Ewing sarcoma is characterized by the presence of a reciprocal chromosomal translocation that gives rise to chimeric transcription factors. These chimeric fusion proteins, which are the main oncogenic drivers in this malignancy [2], involve a member of the FET gene family (*FUS*, *EWSR1*, or *TAF15*) and a transcription factor of the ETS gene family (*FLI1*, *ERG*, *FEV*, *ETV1*, or *ETV4*, among others). The chromosomal translocation t(11;22)(q24;q12) is the most frequent alteration. This generates an aberrant protein constituted, specifically, by the fusion of the N-terminal region of the *EWSR1* gene with the C-terminal region of the ETS transcription factor *FLI1*, where the DNA binding domain resides. As a consequence, the chimeric transcription factor EWSR1::FLI1 modulates the expression, directly or indirectly, of hundreds of genes that globally determine a specific genetic profile [1,2,3]. Thus, EWSR1::FLI1 is placed at the summit of a complex transcriptional regulation network that defines a specific oncogenic expression program.

In the last few years, it has been proposed that there could be two Ewing sarcoma cell populations from a phenotypic point of view that would coexist in Ewing sarcoma tumors [4]. The differences between these two cell populations would be attributable to different expression levels of the aberrant transcription factor. The main population in the tumor would be constituted by tumor cells with high levels of the chimeric protein (EWSR1::FLI1^high^ phenotype) and characterized by elevated proliferation and an undifferentiated cellular status. On the other hand, the minor population, which expresses low levels of EWSR1::FLI1 expression (the EWSR1::FLI1^low^ phenotype), would be characterized by a reduced proliferation rate and increased migration and invasiveness [4,5]. Since EWSR1::FLI1 regulates a specific gene expression profile, these different phenotypes should be the consequence of those genes regulated by EWSR1::FLI1. Thus, deciphering the contribution of specific genes to Ewing sarcoma phenotypes is relevant to understanding Ewing sarcoma biology and identifying pathways involved in proliferation and migration/invasiveness. Our research group has predominantly focused on looking up genes regulated positively or negatively by EWSR1::FLI1 in Ewing sarcoma cells (e.g., NR0B1/DAX1, CCK, LOX, SPRY1, DKK2, or FEZF1) [6,7,8,9,10,11].

In order to identify potential actors involved in cell migration and invasion capability in Ewing sarcoma, we focus, on this occasion, on genes upregulated in the EWSR1::FLI1^low^ phenotype that were also associated with the regulation of these processes. Thus, a RNAseq analysis performed in the cell model A673/TR/shEF, in which EWSR1::FLI1 is downregulated by a specific doxycycline-dependent shRNA [8], showed that *CD44* (Cluster of Differentiation 44) was one of the most overexpressed genes upon EWSR1::FLI1 downregulation. CD44 is a transmembrane protein characterized by an extracellular domain with multiple ligand-binding properties. Hyaluronic acid (HA) has been described as its major ligand among other molecules such as osteopontin, glycosaminoglycans, or collagen [12]. It has been widely accepted that the molecules of the extracellular matrix and its receptors play an essential role in the development of tumors [13]. Interestingly, CD44 protein has been linked to invasion, metastasis, and increased aggressiveness in several solid and hematological malignancies, such as breast cancer, hepatocarcinoma, non-Hodgkin lymphoma, or acute myeloid leukemia, among others [13,14], and it has been identified as a prognostic factor in osteosarcoma, breast cancer, ovarian cancer, gastric cancer, and gallbladder cancer [14,15,16,17,18,19]. Moreover, CD44 is considered a molecular target susceptible to pharmacological therapy [14]. To our knowledge, the role of CD44 in Ewing sarcoma has been superficially elucidated in the context of vasculogenic mimicry, facilitating the formation of vasculogenic structures in the absence of angiogenic factors [20]. In consequence, it would be interesting to define the complete functional spectrum of CD44 in Ewing sarcoma, with a focus on its possible role in cell migration and invasiveness.

In this work, we showed that the CD44 standard isoform (CD44s) is negatively regulated by EWSR1::FLI1 and thus overexpressed in the EWSR1::FLI1^low^ phenotype. The expression of CD44s in three independent Ewing sarcoma cell lines reduced cell proliferation without impairing the clonogenic ability of these cells. More interestingly, CD44s expression increased cell migration and invasiveness in two Ewing sarcoma cell lines (A673 and MHH-ES1). According to these findings, we suggest that CD44s could be involved in the migration/invasion features that characterize the EWSR1::FLI1^low^ phenotype.

## 2. Results

### 2.1. CD44s Is Upregulated upon EWSR1::FLI1 Knockdown

The main objective of this study was to identify genes that were downregulated by EWSR1::FLI1 in Ewing sarcoma that could be potentially relevant for the tumorigenesis process. For this aim, we used the A673/TR/shEF cell line, which is a well-established cell model for EWSR1::FLI1 downregulation [8]. In this Ewing sarcoma cell model, EWSR1::FLI1 is efficiently downregulated upon doxycycline-mediated expression of a specific shRNA designed against the EWSR1::FLI1 chimeric mRNA. In consequence, in the absence of doxycycline, A673 Ewing sarcoma cells produce high levels of EWSR1::FLI1 (EWSR1::FLI1^high^), while in the presence of doxycycline, low levels of EWSR1::FLI1 (EWSR1::FLI1^low^) are achieved. 

Firstly, we analyzed RNAseq data obtained from the A673/TR/shEF cell line under EWSR1::FLI1^high^ and EWSR::FLI1^low^ conditions (Figure 1). To obtain a list of strongly regulated EWSR1::FLI1 target genes, we selected all the genes that showed log_2_ fold changes > 2.5 or < 2.5 and an adjusted *p*-value < 0.01. According to these conditions, a total of 44 genes were downregulated upon EWSR1::FLI1 knockdown (log_2_ fold change < 2.5, adjusted *p*-value < 0.01), thus being characteristic of the EWSR1::FLI1^high^ phenotype; 273 genes were upregulated upon EWSR1::FLI1 knockdown (log_2_ fold change > 2.5, adjusted *p*-value < 0.01), thus being characteristic of the EWSR1::FLI1^low^ phenotype (Figure 1A and Appendix A). 

To further identify genes involved in cell migration, invasion, or metastasis using an unbiased approach, each gene in the dataset was cross-referenced with the PubMed database using the terms invasion, migration, and metastasis. Afterward, the number of published articles for each gene with any of those terms was noted, and then the genes were ranked accordingly (Appendix A). Figure 1B shows the results obtained after this analysis. The vast majority of genes with a higher number of citations, including the terms invasion, migration, or metastasis, belonged to genes upregulated in the EWSR1::FLI1^low^ phenotype. Interestingly, only *CCK*, a gene upregulated by EWSR::FLI1 as previously described by our group, showed a high number of citations in the group of genes upregulated in the EWSR1::FLI1^high^ phenotype [8]. Among all genes strongly regulated by EWSR1::FLI1, *CD44* was the one with the largest number of citations, demonstrating a clear association between this gene and cell migration, invasion, and metastatic processes in the literature. Therefore, we selected this gene for additional analysis as it could play a key role in the EWSR1::FLI1^low^ phenotype.

According to the Ensembl database, there are more than 20 CD44 mRNA isoforms; however, only a few of them have been experimentally elucidated (https://www.ensembl.org/; ID:ENSG00000026508, accessed 20 January 2023). These mRNA isoforms produce different CD44 proteins with different molecular weights, containing specific domains that interact with several proteins in the extracellular matrix [14]. Thus, it is important to know what specific CD44 isoforms are present upon EWSR1::FLI1 knockdown, as this may be relevant to understanding the role of CD44 in Ewing sarcoma. To determine the most overexpressed isoform, we performed a deeper characterization analysis of the RNAseq dataset. Seven CD44 isoforms were detected by using TopHat v2.1.1 software, a fast splice junction mapper used to detect mRNA isoforms in RNAseq analysis (Appendix A). However, the CD44v4 isoform (NM_001001391), also called the CD44 standard isoform (CD44s), was the only one significantly overrepresented (fold change > 10, adjusted *p*-value < 0.0001) upon EWSR1::FLI1 knockdown (Appendix A). The CD44s protein (Uniprot reference: P16070-12) is constituted by nine exons, and it lacks a large region of the ectodomain but still maintains the domains involved in the specific recognition of hyaluronic acid (HA) [14]. CD44s is heavily glycosylated, showing an apparent molecular weight of about 80 KDa in a western blot analysis [12,21,22].

CD44 upregulation upon EWSR1::FLI1 knockdown in A673 cells was confirmed by RT-qPCR (Figure 2A) and western blot analysis (Figure 2B). Cell immunostaining corroborated that CD44 was overexpressed upon EWSR1:FLI1 knockdown, and the protein was localized in the cell membrane (Figure 2C). Next, we analyzed if the low expression of CD44 in A673 cells (EWSR1::FLI1^high^) was also observed in other Ewing sarcoma cells and/or other sarcoma types. As shown in Figure 2D,E, CD44 was not detected in any Ewing sarcoma cell line analyzed (*n* = 7) at the mRNA or protein level, indicating that CD44 is not expressed in Ewing sarcoma cell lines maintained in standard culture conditions (EWSR1::FLI1^high^). Interestingly, CD44 was expressed at high levels in cell lines derived from osteosarcomas (CAL-72, SAOS-2, and U2OS), a chondrosarcoma (CAL-78), and a fibrosarcoma (HT1080) at mRNA and protein levels. To further evaluate if CD44 is highly expressed in sarcomas other than Ewing sarcoma, we examined the expression dataset publicly available in the DEPMAP portal (https://depmap.org/portal/; accessed 15 December 2022). This dataset confirmed the elevated expression of CD44 in other sarcoma subgroups (osteosarcoma, rhabdomyosarcoma, chondrosarcoma, leiomyosarcoma, synovial sarcoma, fibrosarcoma, and liposarcoma) compared to Ewing sarcoma cell lines (Appendix A). 

### 2.2. CD44s Expression Reduces Cell Proliferation without Impairing Clonogenicity

Our data indicate that CD44 expression was upregulated upon EWSR1::FLI1 knockdown, and thus it is expressed in the EWSR1::FLI1^low^ phenotype of Ewing sarcoma cells. Moreover, CD44 seems to play a crucial role in the regulation of cell migration and invasiveness, and it has been shown to be a prognostic marker in several tumors [12,14,16,22,23,24,25]. Hence, we considered studying the role of CD44 as a potential driver for Ewing sarcoma progression and aggressiveness.

To evaluate this hypothesis, we engineered three Ewing sarcoma cell lines with a CD44s expression regulation system (that is, doxycycline-inducible), as it was the isoform preferentially overexpressed in the EWSR1::FLI1^low^ phenotype (Appendix A). Ewing sarcoma cell lines selected for these experiments were representative of the most frequent gene fusions observed in Ewing sarcoma, that is, A673 (EWSR1::FLI1 type 1 fusion, EWSR1 exon 7-FLI1 exon 6), MHH-ES1 (EWSR1::FLI1 type 2 fusion, EWSR1 exon 7-FLI1 exon 5), and CADO-ES1 (EWS::ERG fusion, EWS exon 7-ERG exon 8).

Figure 3 shows the molecular characterization of the cell lines generated. Once stable cell populations were obtained, the upregulation of CD44s upon doxycycline stimulation was molecularly confirmed by RT-qPCR (Figure 3A) and western blot analysis (Figure 3B). On the western blot, CD44s showed an apparent molecular weight of 80 KDa in all cell lines stimulated with doxycycline, indicating that CD44s was adequately glycosylated. Thus, these cell models properly reproduced the molecular characteristics of the CD44 isoform preferentially observed upon EWSR1::FLI1 knockdown in the A673/TR/shEF cell model (Figure 2B and Appendix A). No changes in the mRNA and protein levels of the chimeric proteins were observed (Figure 3A,B), confirming that the upregulation of CD44s does not affect fusion gene expression. In addition, cell immunostaining (Figure 3C) showed that CD44s was mainly localized at the cell membrane, in a similar way to that observed in A673 EWSR1::FLI1^low^ cells. Once these cell models were characterized, we analyzed the effect of CD44s upregulation on cell proliferation, clonogenic capability, cell migration, and invasive properties in order to determine the possible role of CD44 in Ewing sarcoma tumorigenesis. 

Figure 4A shows the effect of CD44 upregulation on cell proliferation in the three Ewing sarcoma cell lines. CD44s expression increased cell duplication time by 26–57% in the three cell lines tested. Thus, the cell doubling time of A673/TR/CD44s in the absence of doxycycline was 38.7 ± 6.25 h, while it increased to 53.77 ± 11.97 h (mean ± SD, *n* = 6 time points analyzed, *p*-value < 0.05, Student’s *t*-test) in the presence of doxycycline (CD44s overexpression). The MHH-ES1/TR/CD44s cell line showed similar behavior; the cell doubling time increased from 45.1 ± 3.48 h to 56.98 ± 5.37 h when CD44s was overexpressed (*n* = 8 time points analyzed, *p*-value < 0.005). Finally, the cell doubling time increased from 33.02 ± 6.86 to 51.97 ± 7.77 in CADO-ES1/TR/CD44s cells expressing CD44s (*n* = 6 time points analyzed, *p*-value < 0.05). Overall, the upregulation of the CD44s isoform in the three Ewing sarcoma cell lines analyzed significantly reduced the cell proliferation rate (Figure 4A).

Next, we performed a clonogenic assay to determine the effect of CD44s expression on the cell’s capability to survive and progress in low-density cell culture conditions. As shown in Figure 4B, the three cell lines showed a significant reduction (range between 40 and 62%) in the amount of dye released after cell de-staining (*p*-value < 0.001, Student’s *t*-test) (Figure 4B). To determine whether this result was due to a decrease in the colony count rather than a reduction in the total cell number, we performed an analysis of the colonies per field (Figure 4C) and their respective areas (Figure 4D). Interestingly, the de-staining differences observed were not attributed to a reduction in the number of individual colonies (Figure 4C), which were similar between doxycycline-stimulated (CD44s^high^) and non-stimulated cells (CD44s^low^), indicating no impairments in clonogenicity. This effect was rather due to a decrease in the colony area (a consequence of the reduction in the number of cells per colony due to the different proliferation rates) (Figure 4D and Appendix A). Taken together, these results suggest that CD44s is involved in cell proliferation without modifying the clonogenic capability of Ewing sarcoma cell lines.

### 2.3. CD44s and LMW-HA Increase Cell Migration

We then performed cell migration assays to investigate whether CD44s overexpression affects the migration of Ewing sarcoma cells. Migration assays, carried out on transwell semi-porous membranes, showed that CD44s upregulation significantly increased the percentage of migrating cells in A673/TR/CD44s and MHH-ES1/TR/CD44s cell lines compared to the control group (Figure 5A). Cell migration of A673 cells expressing CD44s increased around 60% (*p*-value < 0.001, *n* = 3 experiments, Student’s *t*-test), while in MHH-ES1 cells it increased around 130% (*p*-value < 0.0005, *n* = 3 experiments). These results confirmed the potential influence of CD44s in migration processes, which are associated with greater tumor aggressiveness. However, this effect of CD44s on cell migration was not observed in CADO-ES1 cells, and no significant differences were found among control cells and cells expressing CD44s (Figure 5A). Additionally, this cell line showed the lowest migration capacity upon CD44s overexpression. Indeed, only 2.09 ± 0.462% (mean ± SD, *n* = 3 experiments) of CADO-ES1 cells migrated through a semi-porous membrane, while the percentage of migrating cells was 13.25 ± 1.45% for A673 cells and 11.09 ± 1.63% for MHH-ES1 (Figure 5A). These findings suggest that CD44s could be involved in the regulation of cell migration in some subgroups of Ewing sarcoma but not in others.

To complement these studies, we decided to explore the effect of CD44s upregulation on cell migration in the presence of the glycosaminoglycan hyaluronic acid (HA) (Figure 5B). HA is the major ligand of CD44, and, remarkably, the CD44s isoform contains exclusively the extracellular domain that interacts with this molecule. We studied the effect of two different types of HA defined by their molecular weight, which in turn reflects the length of the disaccharide chain: high molecular weight HA (HMW-HA, > 500 kDa) and low molecular weight HA (LMW-HA, < 50 kDa). 

Transwell membranes coated with HMW-HA reduced the percentage of migrating cells by 41–64% in A673/TR/CD44s (*p*-value < 0.001, *n* = 3 experiments, Student’s *t*-test), 84–92% in MHH-ES1/TR/CD44s (*p*-value < 0.001, *n* = 3 experiments), and 50–56% in CADO-ES1/TR/CD44s (*p*-value not significant, *n* = 3 experiments) (Figure 5B). This effect was observed in both unstimulated cells (CD44s^low^) and doxycycline-stimulated cells (CD44s^high^). Thus, HMW-HA not only blocked the positive effect of CD44s upregulation on cell migration observed in A673 and MHH-ES1 cells but even reduced basal levels of cell migration. The fact that HMW-HA was also able to reduce migration in cells without CD44s expression suggests that its negative effect on cell migration was independent of CD44s expression. By contrast, LMW-HA significantly increased the percentage of migrating cells in A673 and MHH-ES1 cells expressing CD44s when compared to uncoated membranes. Interestingly, this effect was only observed in cells expressing CD44s but not in basal conditions (CD44s^low^), suggesting that the effect of LMW-HA on cell migration was specific and dependent on CD44s expression (*p*-value< 0.05, *n* = 3 experiments) (Figure 5B). Again, the behavior of CADO-ES1 was different from that observed in A673 and MHH-ES1 cells. Therefore, LMW-HA did not affect cell migration regardless of whether CD44s was expressed in the CADO-ES1 cell line. 

Overall, CD44s overexpression increased cell migration in two of the Ewing sarcoma cell lines tested (A673 and MHH-ES1). More interestingly, HMW-HA and LMW-HA modulated cell migration in two distinctly opposite ways. HMW-HA reduced basal cell migration in a CD44s-independent way, while LMW-HA increased cell migration only in CD44s-expressing cells.

### 2.4. CD44s Expression Increases Cell Invasiveness

Finally, we aimed to elucidate the role of CD44s overexpression on cell invasion by performing matrigel invasion assays and three-dimensional cell culture assays (tumor spheroids).

To study the effect of CD44s expression on the ability of cells to degrade and migrate through a well-structured extracellular matrix, cells were plated onto matrigel-pretreated semi-porous membranes and allowed to migrate through matrigel for 24 h. As shown in Figure 6A, no significant differences among the control (CD44s^low^) and doxycycline-treated groups (CD44s^high^) were observed in any of the Ewing sarcoma cell lines tested. Since cell invasion was assayed for a period of 24 h in these experiments, we considered the possibility that more time was necessary to detect significant effects. Therefore, we carried out long-term experiments using three-dimensional tumor spheroids embedded in a collagen matrix.

A673, MHH-ES1, or CADO-ES1/TR/CD44s cells were resuspended in small droplets of culture medium, and tumor spheroids were generated by a gravity-mediated aggregation method. Once cell aggregation was achieved, tumor spheroids were embedded in a collagen type I matrix for 3 days and subsequently monitored for 6–7 days by light microscopy to determine the cell invasiveness of the surrounding matrix. To quantify the area invaded by the cells, two regions called “core” and “invasion area”, were defined, and their areas were quantified from the images obtained from each individual spheroid (Figure 6B). The “core” was defined as the region delimited by the major mass of compacted cells. This nucleus contains cells that are actively proliferating. The “invasion area” was defined as the surface invaded by the cells that colonize the surrounding matrix by degrading and infiltrating the collagen matrix. After 144 hours, core areas were smaller in doxycycline-treated cells (CD44s^high^) compared to untreated cells (CD44s^low^) in the three Ewing sarcoma cell lines analyzed (Figure 6C). This finding indicates that CD44s^high^ cells proliferate at a slower rate, in accordance with the previous results. By contrast, the invasion area of A673 and MHH-ES1 cells expressing CD44s (CD44s^high^) significantly increased compared to CD44s^low^ spheroids (Figure 6D). CADO-ES1 spheroids did not show significant differences between CD44s^high^ and CD44s^low^ according to the cell migration results described above. As expected, this cell line showed the lowest invasiveness of the three Ewing sarcoma cell lines studied.

Based on these results, CD44s expression increases the invasiveness of some Ewing sarcoma cell lines. Altogether, CD44s expression is able to increase cell migration and invasiveness of Ewing sarcoma cells, which may in turn contribute to the malignant phenotype of these tumors. Our findings suggest that CD44s could be a relevant gene in the Ewing sarcoma EWSR1::FLI1^low^ phenotype, contributing to the migratory and invasive features that characterize this phenotype.

## 3. Discussion

Ewing sarcomas are molecularly characterized by pathognomonic chromosomal translocations that result in the production of chimeric transcription factors [1]. Recently, it has been proposed that fusion protein levels can fluctuate in individual tumor cells, generating specific genetic profiles that are associated with EWSR1::FLI1 high and low levels and that in turn are associated with different cell behaviors [4,5]. In this work, we have focused on the gene expression profile expressed in the EWSR1::FLI1^low^ phenotype in order to identify genes that could contribute functionally to the cell migration and invasion phenotype characteristic of EWSR1::FLI1^low^ cells.

By using the extensively studied A673/TR/shEF cell line [8], we showed that CD44 was dramatically upregulated upon EWSR1::FLI1 downregulation (that is, in the EWSR1::FLI1^low^ phenotype), suggesting that CD44 is repressed by EWSR1::FLI1 in EWSR1::FLI1^high^ Ewing sarcoma cells. According to this, CD44 expression levels were undetectable in all Ewing sarcoma cells analyzed but were moderately or highly expressed in a variety of sarcoma cells from other origins, suggesting that repression of CD44 expression in EWSR1::FLI1^high^ Ewing sarcoma cells is a common feature of this tumor. The mechanism potentially involved in the negative regulation of CD44 expression in Ewing sarcoma under these conditions remains unknown, but it clearly correlates with the expression levels of EWSR1::FLI1 protein. 

CD44 is a well-established pro-metastatic gene in several malignancies [13,14,16,24], including some sarcomas [14,26,27,28,29,30]. The available data suggest that CD44 plays a relevant role in the regulation of cell migration/cell invasion processes associated with tumor malignancy. Interestingly, the role of CD44 in Ewing sarcoma has been superficially explored. Only one study on Ewing sarcoma was performed by Paulis et al. [20], focusing on two different Ewing sarcoma cell lines: EW7 and SIM/EW27. The EW7 cell line, which is more aggressive, presented higher levels of CD44 in the expression array compared to the less aggressive SIM/EW27 cell line. The in vitro results showed that high levels of CD44 in the EW7 cell line were related to increased vasculogenic mimicry (the ability to transdifferentiate into cells with endothelial features and thus form vasculogenic networks) and a faster and more efficient adherence to immobilized HA. The CD44 isoform characterization performed on these cells showed that there were significant differences in most isoforms expressed among the two cell lines (including CD44s). Alternatively, knockdown experiments of CD44 in these cell lines demonstrated that its expression is necessary for vasculogenic network formation and cell migration.

In our study, in order to identify the isoforms preferentially upregulated upon EWSR1::FLI1 knockdown in Ewing sarcoma A673 cells (that is, in the EWSR1::FLI1^low^ phenotype), we analyzed in detail our RNAseq dataset. Thus, we could determine that the isoform preferentially upregulated upon EWSR1::FLI1 knockdown was the denominated CD44 standard isoform (CD44s) (Appendix A). Interestingly, this isoform retains the domain that binds to HA but lacks the regions involved in the interaction of CD44 with other extracellular proteins such as proteoglycans, collagen, osteopontin, collagen, or fibronectin. 

We generated three Ewing sarcoma cell lines (A673/TR/CD44s, MHH-ES1/TR/CD44s, and CADO-ES1/TR/CD44s) to study the functional role of CD44 in Ewing sarcoma. In these cell models, CD44s expression was tightly controlled by a doxycycline-inducible expression system. Although the CD44s isoform has a theoretical molecular weight of 39.41 kDa (Uniprot ID: P16070-12), we observed that in the three cell lines it migrated up to 80–95 kDa on a western blot, which is consistent with extensive glycosylation as previously described [21,22].

CD44s upregulation also reduced cell proliferation in the three Ewing sarcoma cells analyzed but did not impair their clonogenic capability. In the absence of CD44s, colonies were thicker, while colonies overexpressing CD44s presented less cell density (Appendix A). Interestingly, the negative effect of CD44s on cell proliferation was observed even in the presence of EWSR1::FLI1 (Figure 3A,B), suggesting that CD44s expression has a remarkable effect on the phenotype of these cells.

Using transwell migration assays, we determined that CD44s expression increased cell migration in A673 and MHH-ES1 Ewing sarcoma cell lines but not in CADO-ES1 (Figure 5A). As we mentioned above, the migration implications of tumor cells with high CD44 levels have been broadly studied [13,14,24,31]. For example, it has been shown that CD44 inhibition in the U251 glioblastoma cell line reduced migration and invasion ability in transwell assays, while its re-expression re-established the high migratory and invasive phenotype [32]. Similar results have been obtained in breast cancer (MDA-MB-231) and osteosarcoma (MG-63 and U2OS) cell lines; CD44 knockout or CD44 silencing in these cell lines (which express high CD44 levels on basal conditions) drastically reduced the migration and invasion rate compared with wild-type cells [25,33]. Consistently, we proposed that CD44s could play a relevant role in the regulation of cell migration in some Ewing sarcoma cell lines. 

As CD44s maintains the extracellular regions necessary to interact with the extracellular component HA, we analyzed how HA affected CD44-induced cell migration (Figure 5B). Interestingly, we observed different effects on cell migration depending on the range of HA molecular weight. The effect of HMW-HA blocking cell migration suggests that HMW-HA blocks cell migration in a dual mechanism, one independent of CD44 and the other probably CD44-dependent due to the more drastic reduction in CD44^high^ cells. Research focused on CD44 in other malignancies has elucidated several mechanisms through which this migration inhibition is fulfilled: (i) HMW-HA decreases the permeability of extracellular matrix (ECM) by strengthening cell-cell junctions, thus preventing invasion (CD44-independent mechanism) [34,35]. (ii) HMW-HA blocks cell migration by enhancing the co-association between NF2 (also known as Merlin protein) and CD44, which activates the tumor suppressor properties of CD44 (CD44-dependent mechanism) [35,36]. (iii) CD44-HA binding for HMW-HA species (over 262 kDa) was determined to be irreversible and to reduce cell surface CD44 clustering, producing impairments in adhesion and signaling processes (CD44-dependent mechanism) [37,38], thus demonstrating a potential anti-metastatic effect. 

By contrast, LMW-HA increases cell migration in A673 and MHH-ES1 cells expressing CD44s but not in control cells that do not express CD44. As above, this effect was not observed in CADO-ES1 cells. Several studies have reported that LMW-HA increases tumor cell migration and invasiveness [35,39]. Koyama et al. demonstrated that cell migration stimulated by LMW-HA is often associated with an epithelial-mesenchymal transition (EMT) mediated by CD44s [40]. In consequence, the microenvironment providing HA could increase or diminish the effect of the axis HA-CD44s, modulating cell plasticity and EMT processes. The major niche where Ewing sarcoma arises is the bone marrow environment [1], which is composed of both stromal and hematopoietic cells. These cells are known to produce abundant HA that regulates cell proliferation/differentiation/survival/migration processes [41,42,43]. Subsequently, we suggest that the microenvironment generating hyaluronan molecules could modulate the Ewing sarcoma cells’ aggressiveness through their interaction with the HA-CD44s axis. This relevant finding has implications for complex biological systems. In neuroblastoma xenograft mouse models, it has been demonstrated that HA, predominantly detected in the periarteriolar space in mouse lungs, co-localized with CD44s-overexpressing cells [44,45,46]. Therefore, the organ tropism to the lungs and the location of the highly expressive CD44s metastatic cells should be further studied in animal models because of the potential influence of HA-CD44s in other migration/metastasis processes such as the homing of the invasive cells.

We have also shown that CD44s expression increases cell invasion in A673 and MHH-ES1 cells using a 3D-based cell culture model (Figure 6D). This experimental approach integrates features of the in vivo tumor behavior and microenvironment, providing advantages compared to the classical 2D-based models [47]. Thus, this methodology allowed us to mimic the characteristics of the tumor mass and the presence of the extracellular matrix components. Our results in A673 and MHH-ES1 cell lines support the influence of CD44 in the invasion ability previously described in other malignancies. For example, positive CD44 head and neck squamous cell carcinoma spheroids showed an increase of 340–750% in invasion compared with negative CD44, which additionally showed resistance to radiotherapy [48].

Our findings support the sequential process underlying the initiation of the pro-migratory/invasive phenotype in EWSR1::FLI1^low^ cells. This process involves upregulation of CD44 expression due to the reduction in EWSR1::FLI1 levels. CD44, which interacts with hyaluronic acid (HA), commonly found in connective tissue and bone marrow, could activate the CD44-HA axis in Ewing sarcoma cells. This axis plays a crucial role in ECM degradation processes mediated by metalloproteases, as well as in cell migration and invasion [35,39]. The interaction between CD44 and HA has been shown to trigger cytoskeletal remodeling and cell signaling changes in the tumor cell, promoting epithelial-mesenchymal transition mechanisms (EMT) [49,50]. These mechanisms enable cells to colonize adjacent or distant tissues. 

CD44 overexpression in the EWSR1::FLI1^low^ cell population could thus participate in a complex network of mechanisms necessary to manifest the aggressive phenotype observed in these cells. In this sense, the analysis of the expression pattern in the A673/TR/shEF cells revealed the upregulation of several genes associated with pro-metastatic phenotypes (Appendix A). Notably, significant upregulation was observed in several metalloprotease genes, including MMP2, MMP10, and MMP13. These results are particularly relevant as metalloproteinases have been described as participating cooperatively with CD44 in the degradation of the ECM components. This finding would support an additive or synergetic activity of the CD44-metalloprotease-ECM degradation axis, constituting a promising target for therapeutical approaches [14].

Of the three Ewing sarcoma cell lines tested, CADO-ES1 showed no differences in their capability of cell migration (Figure 5) or invasion (Figure 6), regardless of CD44s expression. One possible explanation for this discrepancy could be that CADO-ES1 was the only cell line with a different chromosomal translocation, which generates the EWSR1::ERG chimeric transcription factor instead of EWSR1::FLI1 (A673 and MHH-ES1 cell lines) [1,51]. Although EWSR1::FLI1 and EWSR1::ERG chimeric proteins have been shown to possess similar transcription activities [52], it is conceivable that there may be differences in their regulatory effects on individual genes. This chimeric protein would define a specific genetic and expression profile or pattern that determines the cell line’s behavior. Interestingly, this cell line has been reported to differentiate spontaneously into neural and mesenchymal cell lineages [53,54,55], and it can acquire chondro-phenotype characteristics in vivo [56]. These findings indicate that CADO-ES1 may have some specific characteristics that make it unique among Ewing sarcoma cell lines. We suggest that these features could determine the different responses observed in CADO-ES1 in relation to CD44 expression. 

While our study has provided valuable insights into the functional implications of CD44 in the context of Ewing sarcoma, it is important to acknowledge the inherent limitations associated with the in vitro models. Therefore, further investigations using more complex systems, such as animal models or patient-derived samples, are required to validate and expand upon these findings. 

To our knowledge, this is the first time CD44s has been reported as a potential driver of cell migration and invasion in Ewing sarcoma. Furthermore, elucidating the specific mechanistic pathway underlying CD44-mediated effects on migration, extracellular matrix degradation, and EMT is essential for identifying potential translational opportunities and therapeutic targets (such as antibodies, HA-oligomers, and aptamers, among others). This could elucidate how CD44s is interconnected with the rest of the elements in a complex cell and genetic profile, determined by the fluctuant levels of the chimeric proteins.

## 4. Materials and Methods

### 4.1. Cell Lines

A673/TR/shEF cells, which express a specific shRNA inducible by doxycycline directed against EWSR1::FLI1 mRNA, have been previously described in detail [8]. A673/TR/shEF cells were maintained in DMEM supplemented with 10% tetracycline-free FBS (Capricorn Scientific, Ebsdorfergrund, Germany), 50 U/mL penicillin, 100 μg/mL zeocin, and 5 μg/mL blasticidin. A673/TR/shEF cells were stimulated with doxycycline (1 µg/mL) (Formedium, Norfolk, UK) to induce the expression of EWSR1::FLI1-specific shRNA. A673 (ATTC; CRL-1598), HT1080 (ATTC; CCL-121), U2-OS (DSMZ; ACC 832), and SAOS-2 (DSMZ; ACC 243) cell lines were cultivated in DMEM. The SK-N-MC (ATCC; HTB-10) cell line was maintained in DMEM supplemented with 1× MEM non-essential amino acids. CADO-ES-1 (DSMZ; ACC 255), MHH-ES-1 (DSMZ; ACC 167), A4573 (RRID: CVCL_6245), and CAL-78 (DSMZ; ACC 449) cell lines were maintained in RPMI 1640 medium. TC-71 (DSMZ; ACC 516) was cultured in IMDM and SK-ES-1 in McCoy’s medium. The CAL-72 (DSMZ; ACC 439) cell line was maintained in DMEM supplemented with 1× insulin transferrin sodium selenite (Merck Life Science). All media were supplemented with 10–20% FBS and penicillin and streptomycin. All cells were periodically tested for mycoplasma contamination (Mycoalert mycoplasma detection kit, #LT07-318, Lonza, Basel, Switzerland) and authenticated by STR profiling at the Genomic Facility at the Biomedical Research Institute (IIB-CSIC, Madrid, Spain).

### 4.2. Establishment of Ewing Sarcoma Cell Lines Expressing the Doxycycline-Inducible CD44s Protein

Ewing sarcoma cell lines A673, MMH-ES-1, and CADO-ES1 were infected with lentiviruses containing the pLenti6/TR expression plasmid (Invitrogen, Waltham, MA, USA) to establish stable cell lines constitutively expressing the tetracycline repressor (TR). One clone for each cell line expressing the highest levels of tetracycline repressor, as assayed by western blot (designed A673/TR, CADO-ES1/TR, and MHH-ES-1/TR), was chosen for the next steps. cDNA corresponding to the CD44s isoform (CCDS31457.1; transcript ID: ENST00000263398.11; Uniprot: P16070-12) was obtained by RT-PCR using total mRNA from A673/TR/shEF stimulated with doxycycline and the following oligonucleotides: CD44s-F (forward) 5′-GGGGTCGACTCCGGACACCATGGACAAGT-3′ and CD44s-R (reverse) 5′-CGGCGGCCGCTCACTATTACACCCCAATCTTCATGT-3′. The amplified fragment was digested with SalI ad NotI, cloned into the pENTR2B plasmid, and transferred by recombination to the lentiviral doxycycline-inducible plasmid pLenti4/TO/V5-DEST. Lentiviral vectors containing the CD44s cDNA were generated with the ViraPower™ Lentiviral Packaging Mix (Thermo Fisher Scientific, Waltham, MA, USA). Subsequently, A673/TR, MHH-ES-1/TR, and CADO-ES1/TR cells were infected with lentiviral supernatant, maintained in standard culture conditions for 24 h to allow recovery, and then selected with zeocin (100 µg/mL). Stable clones were checked for CD44s expression by RT-qPCR and western blot after 72 h of stimulation with doxycycline (1 µg/mL). Clones showing medium-high levels of protein induction upon doxycycline stimulation were selected for additional studies.

### 4.3. Quantitative RT-PCR (RT-qPCR)

Reverse transcription and quantitative PCR conditions, primers, and TaqMan probe sequences specific for EWSR1::FLI1 and TBP were described elsewhere [8,9]. The TaqMan probe for CD44 (hs01075862_m1) was purchased from Life Technologies (San Diego, CA, USA). A real-time quantitative PCR was performed with SYBR Green to analyze the EWSR1::ERG expression levels in the CADO-ES1 cell line. The primer sequences were the following: EWSR1::ERG-Forward 5′-TCCTACAGCCAAGCTCCAAGTC-3′and EWSR1::ERG-Reverse 5′-TGTTGGGTTTGCTCTTCCGCTC-3′. Reactions were run on a RotorGene 6000 (Qiagen, Hilden, Germany). The cycle threshold (Ct) for each gene and TBP was calculated using the RotorGene software (v. 2.3.1). Relative expression for each gene was calculated as 2^−∆Ct^, where ∆Ct = Ct_gene_ − Ct_TBP_.

### 4.4. Western Blot Analysis and Antibodies

The western blot is described in detail elsewhere [9]. Primary antibodies were as follows: anti-FLI1 rabbit monoclonal antibody (#ab133485), anti-CD44 rabbit monoclonal antibody (#ab157107, against the intracellular domain of CD44), and HRP-anti-α-tubulin antibody (#ab185067), all purchased from Abcam (Cambridge, UK). Mouse anti-rabbit IgG (#sc-2357) horseradish peroxidase-conjugated secondary antibody was purchased from Santa Cruz Biotechnology (Dallas, TX, USA).

### 4.5. Immunofluorescence

A total of 4000–5000 cells were seeded on glass-bottomed 96-well plates. After 24 h, cells were stimulated with doxycycline (1 µg/mL) for 72 h, and then the growth medium was removed and cells were washed in PBS. After that, cells were fixed for 10 min in 4% paraformaldehyde, permeabilized with 0.1% Triton X-100 in PBS, washed with PBS, and incubated overnight at 4 °C with anti-CD44 primary mouse antibody (#ab6124) diluted in PBS supplemented with 5% FBS. Next, cells were washed 4 × 5 min with PBS and incubated for 1 h at room temperature with a secondary anti-mouse antibody conjugated with Alexa Fluor 594 (#ab150116) diluted in PBS supplemented with 5% FBS. Cells were counterstained for 4 min with the nuclear stain 4,6-diamidino-2-phenylindole (DAPI; 0.1–0.2 µg/mL) and washed 4 × 5 min with PBS. Cells were visualized and photographed under a fluorescence microscope (Leica, Wetzlar, Germany).

### 4.6. Cell Proliferation Assay 

Cells were maintained in culture medium supplemented with 10% tetracycline-free FBS, with or without doxycycline (1 μg/mL), for 30–34 days. When cells reached 70–80% confluence, they were trypsinized, counted, and re-plated in a new p100 plate, which was maintained in this way until completion after 30–34 days. The culture medium containing fresh doxycycline was changed every 3 days. The number of population doublings was calculated with the formula: n° of population doublings = log2 (number of cells at the initial time/number of cells at the final time). Cell duplication time was calculated in each cycle of cell seeding–trypsinization as cell doubling time = time elapsed between cycles/n° cell population doubling observed in this period.

### 4.7. Clonogenic Assay 

A673/TR/LUC/CD44s, MMH-ES-1/TR/CD44s, and CADO-ES-1/TR/CD44s cells were plated in triplicate in a 12-multiwell plate at 1 × 10^3^, 5 × 10^3^, or 3 × 10^3^ cells per well. Thereafter, they were treated with or without doxycycline (1 μg/mL) for 9–10 days in complete culture medium. The culture medium containing fresh doxycycline, where appropriate, was changed every 3 days. Finally, colonies were fixed with formaldehyde (8% mass/volume) for 8 min, stained with crystal violet (0.5% aqueous solution) for 2 min, and photographed. Cells were de-stained using 50% ethanol and 0.1 M sodium citrate at pH 5.2. Absorbance was quantified at 540 nm using an Infinite M200 microplate reader (Tecan). Five random fields from each well were analyzed by two independent researchers. Finally, the number of colonies per field was calculated. Colony area analysis has been described elsewhere [57].

### 4.8. Transwell Migration and Invasion Assay

Cells were maintained in culture medium supplemented with 10% tetracycline-free FBS, with or without doxycycline (1 μg/mL), for 48 h. After 48 h of pre-stimulation, a total of 2.5 × 10^5^ Ewing sarcoma cells (A673, MHH-ES-1, and CADO-ES1) or 10^4^ HT1080 cells were seeded on the pretreated (or control non-treated) insert membrane (12-multiwell plates, PET membrane, 8 µm pore; Corning, New York City, NY, USA). Insert membranes were pretreated overnight with matrigel (250 µg/mL; at room temperature; ref: 356234; Corning, NY, USA), high molecular weight hyaluronic acid (HMW-HA, >500 KDa; 2 mg/mL; at 4 °C), and low molecular weight hyaluronic acid (LMW-HA, <50 KDa; 2 mg/mL; at 4 °C). Cells were allowed to migrate for 24 h. Subsequently, the membranes were removed from the insert and gently scraped the upper side with a cotton swab; migrating cells present on the bottom side were fixed with formaldehyde (8% mass/volume) for 8 min, stained with crystal violet (0.5% aqueous solution) for 2 min, washed gently with distilled water 3 × 2 min, and photographed with a Brightfield microscope (Leica, Wetzlar, Germany). Five random fields from each membrane were analyzed by two independent researchers. Finally, the percentage of migration was calculated (number of migrated cells/cells seeded).

### 4.9. Spheroid Invasion Assay

Multicellular spheroids of Ewing sarcoma cell lines were prepared by the hanging droplet method as previously described [58,59]. A total of 5 × 10^3^ cells in a 20-μL droplet of culture medium, supplemented with 10% FBS, were allowed to aggregate by gravity for three days. Afterwards, spheroids were collected and resuspended in collagen I rat tail gel matrix (final concentration 1.5–2 mg/mL; ref: A1048301; Thermo Fisher Scientific, Waltham, MA, USA) onto 12-multiwell plates. Thereafter, spheroids were treated with or without doxycycline (1 μg/mL) for 7 days in complete culture medium. The culture medium containing fresh doxycycline, where appropriate, was changed every 3 days. The spheroids were monitored and photographed at 4× magnification with an EVOS™ Digital Color Fluorescence Microscope (Thermo Fisher Scientific, Waltham, MA, USA). Images were analyzed with ImageJ software (v1.52).

### 4.10. Transcriptomic Analysis (RNAseq)

RNA was extracted using TRI-REAGENT according to the manufacturer’s protocol (Sigma-Aldrich, Saint Louis, MO, USA) and additionally purified using the RNeasy Mini Elute Cleanup Kit (Qiagen). The mRNA library was obtained with the ScriptSeq v2 RNA-seq library preparation kit (Epicentre, Catalogue number ssv21124), and paired-end sequencing (2 × 75 pb) was carried out in a NextSeq 550 (Illumina). The number of genes covered at 30× was 6086. The average percentage of aligned reads to a single location in the reference genome ranged from 70.08% to 70.24%. Data analysis was carried out using RNA-Seq Alignment (BaseSpace Workflow) v1.1.0. Briefly, mapping was performed with STAR v2.5.0, and transcript abundance was performed with Cufflinks v2.2.1, using GRCh37 as the transcriptome reference. Differences in expressed features from quantification tables were calculated with DESeq2 (v1.6.3). Genes whose *p*-value was < 0.01 were considered differentially expressed between two experimental conditions. 

RNAseq data have been deposited in NCBI’s Gene Expression Omnibus [60] and are accessible through GEO Series accession number GSE232739 (https://www.ncbi.nlm.nih.gov/geo/query/acc.cgi?acc=GSE232739, updated 22 May 2023).

### 4.11. Pubmed Terms Search

The NCBI eutils tool was used to find citations where each of the differentially expressed genes appeared along with the terms: metastasis, invasion, or migration. A specific Python script was developed in which the search was split into batches to ease computational processing. Metastasis, invasion, and migration terms were searched separately and in combination (by using the search condition: METASTASIS OR INVASION OR MIGRATION). The custom script and command line can be found in this github repository: https://github.com/BU-ISCIII/eutils_search_cd44_paper; updated 26 May 2023.

### 4.12. Statistical Analysis 

For a single comparison of two groups, a two-tailed Student’s *t*-test was used. All statistical calculations were made using the GraphPad Prism statistical software version 8.3.0 (GraphPad Software, San Diego, CA, USA).

## 5. Conclusions

CD44s is upregulated upon EWSR1::FLI1 knockdown in the A673 Ewing sarcoma cells. This transmembrane protein increases cell migration (in A673 and MHH-ES1 cell lines) and reduces cell proliferation in Ewing sarcoma cells (in A673, MHH-ES1, and CADO-ES1 cell lines). Hyaluronic acid, the major CD44 ligand, can interact with CD44s and modulate cell behavior depending on its molecular weight (MW). Thus, while HMW-HA blocks cell migration, LMW-HA increases it in a CD44s-dependent way. In addition, CD44s expression induced invasiveness in tumor spheroids. Nevertheless, further research is needed to determine the underlying mechanisms interconnecting the CD44s-HA axis and its crosstalk with other elements of the EWSR1::FLI1^low^ cell profile.

## Figures and Tables

**Figure 1 ijms-24-11774-f001:**
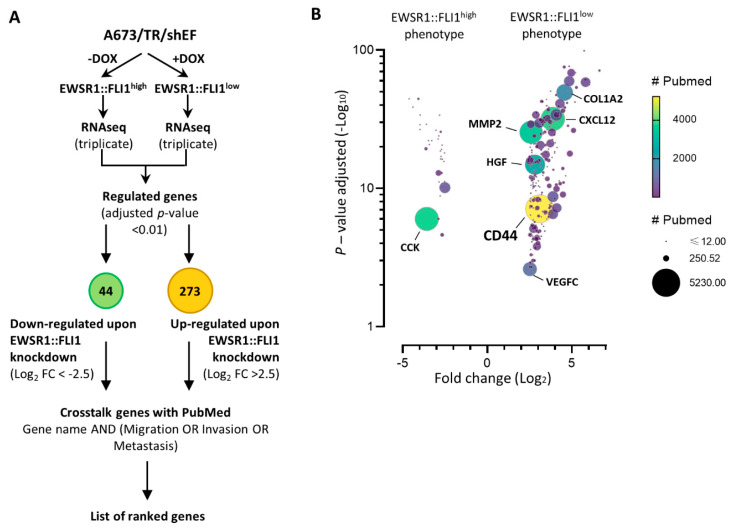
Genes regulated by EWSR1::FLI1 in Ewing sarcoma cell line A673. (**A**) A673/TR/shEF cells were cultured in the absence or presence of doxycycline (DOX, 1 µg/mL, 72 h) and analyzed by RNAseq to identify EWSR1::FLI1-regulated genes. (**B**) Genes regulated by EWSR1::FLI1 were cross-referenced with the PubMed database using the terms invasion, migration, and metastasis. The graph represents the fold change and adjusted *p*-value for each upregulated and downregulated gene and the number of citations for each gene with these terms (which is proportional to the circle areas).

**Figure 2 ijms-24-11774-f002:**
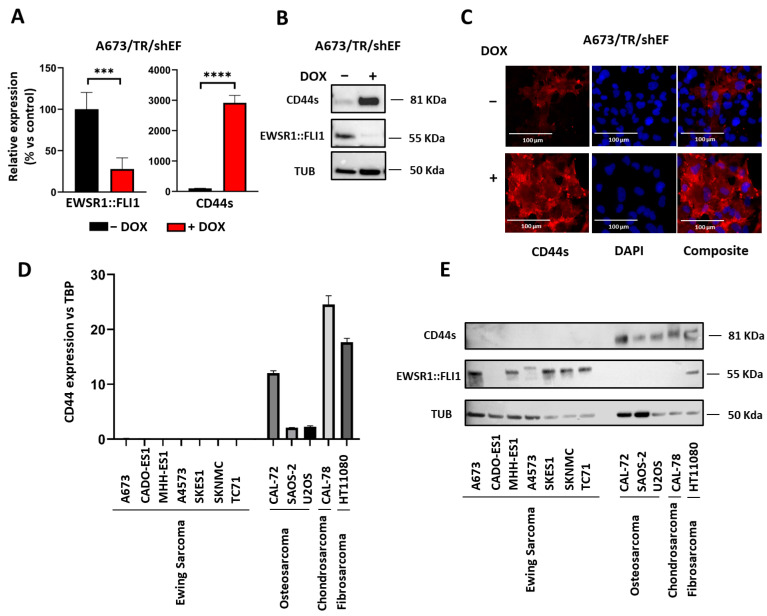
CD44s expression correlates negatively with EWSR1::FLI1 levels in Ewing sarcoma cell line A673 and is not expressed in other Ewing sarcoma cell lines (EWSR1::FLI1^high^ phenotype). (**A**) CD44s mRNA expression (RT-qPCR) in A673/TR/shEF cell cultures in the absence (EWSR1::FLI1^high^) or presence (EWSR1::FLI1^low^) of doxycycline (DOX 1 µg/mL, 72 h). CD44s mRNA levels impressively increased upon EWSR1::FLI1 knockdown (mean ± SD, three experiments performed in triplicate) (*** *p* < 0.001, **** *p* < 0.0001; Student’s *t*-test). (**B**) Western blot analysis confirmed the upregulation of CD44s protein upon EWSR1::FLI1 knockdown. (**C**) A673/TR/shEF cells were cultured in the absence or presence of doxycycline, and CD44s protein was detected by immunofluorescence. CD44s is located in the extracellular membrane, and its expression is notably increased when cells are stimulated with doxycycline (Scale bar: 100 µm). (**D**) CD44s mRNA expression (RT-qPCR) in several Ewing sarcoma, osteosarcoma, chondrosarcoma, and fibrosarcoma cell lines (mean ± SD). (**E**) Western blot analysis of CD44s and EWSR1::FLI1 in several Ewing sarcoma, osteosarcoma, chondrosarcoma, and fibrosarcoma cell lines. CD44s was not detected in any of the Ewing sarcoma cell lines tested, which expressed high levels of the EWSR1::FLI1 protein. By contrast, CD44 was expressed in cell lines derived from other sarcoma types, both at the mRNA (**D**) and protein levels (**E**). CADO-ES1 is an Ewing sarcoma cell line that expresses the chimeric EWSR1::ERG protein instead of EWSR1::FLI1, which is not recognized by the anti-FLI1 antibody used.

**Figure 3 ijms-24-11774-f003:**
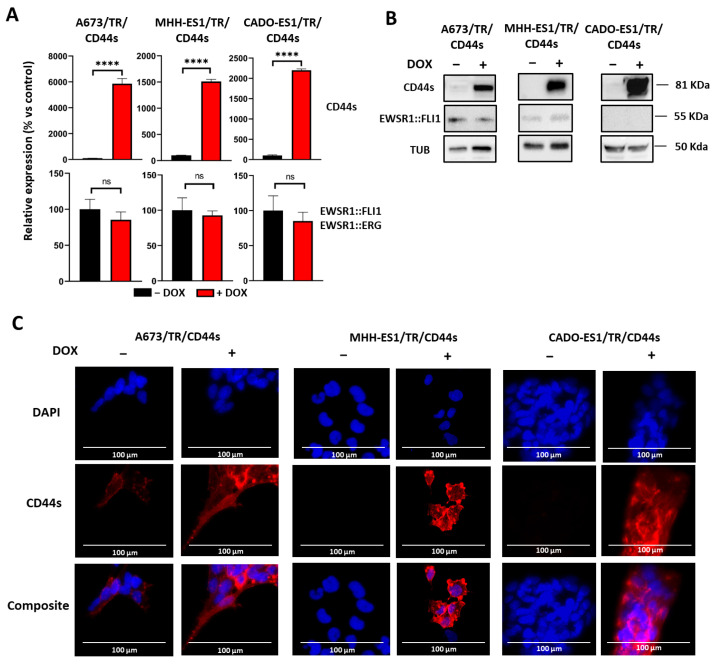
Characterization of Ewing sarcoma cell models generated to study the role of CD44s in Ewing sarcoma. A673, MHH-ES1, and CADO-ES1 Ewing sarcoma cell lines were engineered to express the CD44s isoform through the control of a doxycycline-inducible system. (**A**) mRNA expression levels (RT-qPCR) of CD44s and EWSR1::FLI1 (A673, MHH-ES1) or EWSR1::ERG (CADO-ES1) upon stimulation with doxycycline (DOX, 1 µg/mL, 72 h) (mean ± SD, three experiments performed in triplicate); **** *p* < 0.0001, ns: not significant, Student’s *t*-test. (**B**) Western blot analysis of CD44s and EWSR1::FLI1 in the same cell lines confirms the upregulation of CD44s upon doxycycline stimulation. EWSR1::FLI1 protein levels were not affected by CD44s overexpression. EWSR1::ERG fusion protein is not recognized by the anti-FLI1 antibody used to detect EWSR1::FLI1. (**C**) The three Ewing sarcoma cell lines were cultured in the absence or presence of doxycycline, and CD44 protein was detected by immunofluorescence. CD44s is located in the cell membrane, and its expression is remarkably increased when cells are stimulated with doxycycline (Scale bar: 100 µm).

**Figure 4 ijms-24-11774-f004:**
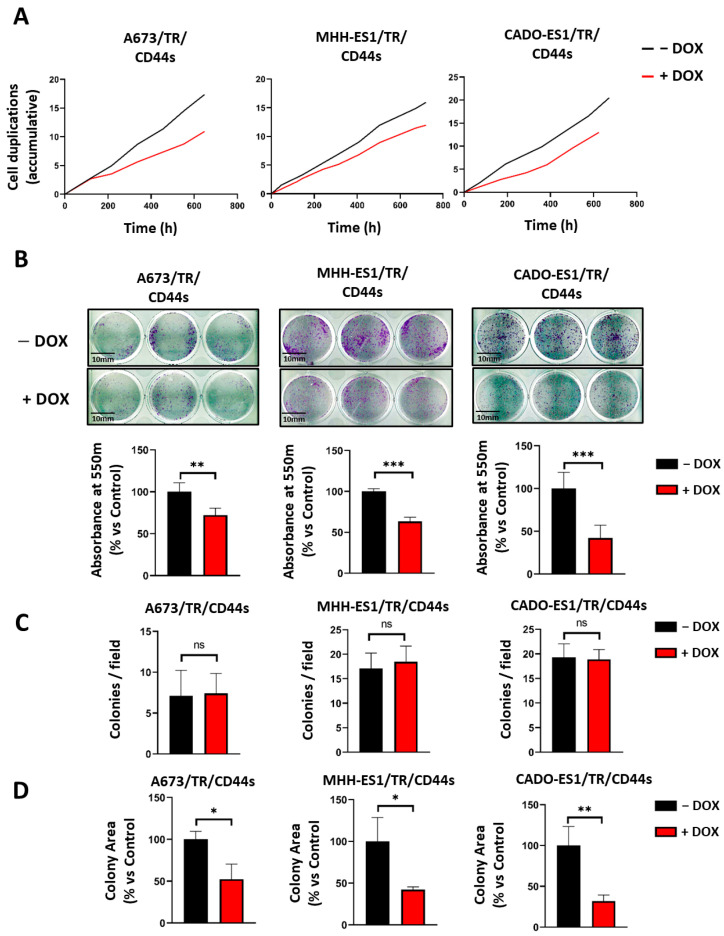
CD44s overexpression reduced Ewing sarcoma cell proliferation but not clonogenic ability. (**A**) Number of cell duplications accumulated over a 30-day period in which cells were cultured in the absence or presence of doxycycline (the graph shows one representative experiment out of two independent experiments performed). CD44s overexpression reduced the duplication time in the three Ewing sarcoma cell lines analyzed. (**B**) Clonogenic assays. CD44s upregulation produced a significant reduction in the number of cells, as quantified by crystal violet de-staining (Scale bar: 10 mm). The graph shows one representative experiment out of three independent experiments performed in triplicate (mean ± SD; ** *p* < 0.01, *** *p* < 0.001, Student’s *t*-test). (**C**) The number of colonies per field was not altered by CD44s upregulation (mean ± SD, ns: not significant, Student’s *t*-test). (**D**) CD44s upregulation decreased colony area quantification (normalized mean ± SD; ns: not significant, * *p* < 0.05, ** *p* < 0.01, Student’s *t*-test).

**Figure 5 ijms-24-11774-f005:**
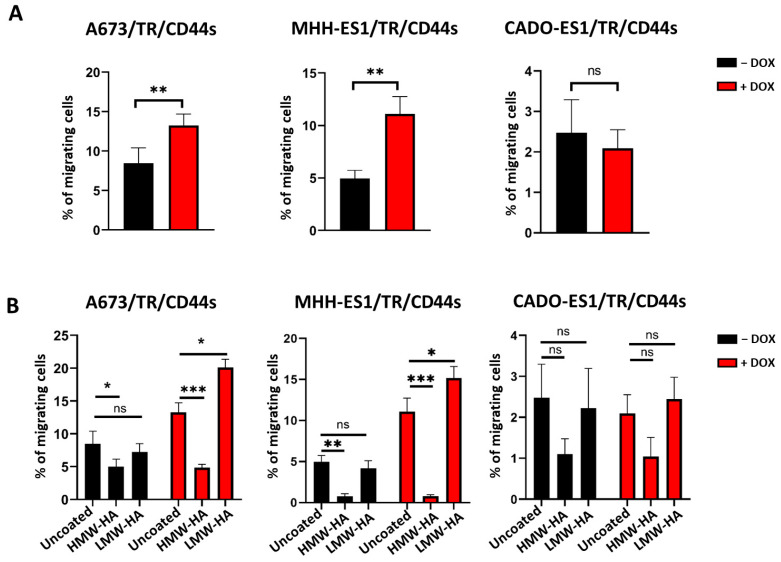
CD44s expression increases cell migration in Ewing sarcoma cell lines. (**A**) Transwell migration assays. Ewing sarcoma cell lines were cultured in the absence or presence of doxycycline to induce CD44s expression, seeded on semi-porous membranes, and the percentage of migrating cells was quantified after 24 h (mean ± SD, three experiments performed in triplicate; ns: not significant, ** *p* < 0.01, Student’s *t*-test). Cell migration was significantly increased in A673/TR/CD44s and MHH-ES1/TR/CD44s cell lines upon CD44s expression. (**B**) Effect of HMW-HA and LMW-HA on cell migration. Cells were seeded on semi-porous membranes uncoated or coated with HMW-HA or LMW-HA, and the percentage of migrating cells was quantified after 24 h. HMW-HA significantly inhibited cell migration of A673, MHH-ES1, and CADO-ES1 cell lines, regardless of whether CD44s was expressed. By contrast, LMW-HA increased cell migration in A673 and MHH-ES1 cells expressing CD44s but not in the CADO-ES1 cell line (mean ± SD, three experiments performed in triplicate; ns: not significant, * *p* < 0.05, ** *p* < 0.01 ***; *p* < 0.001, Student’s *t*-test).

**Figure 6 ijms-24-11774-f006:**
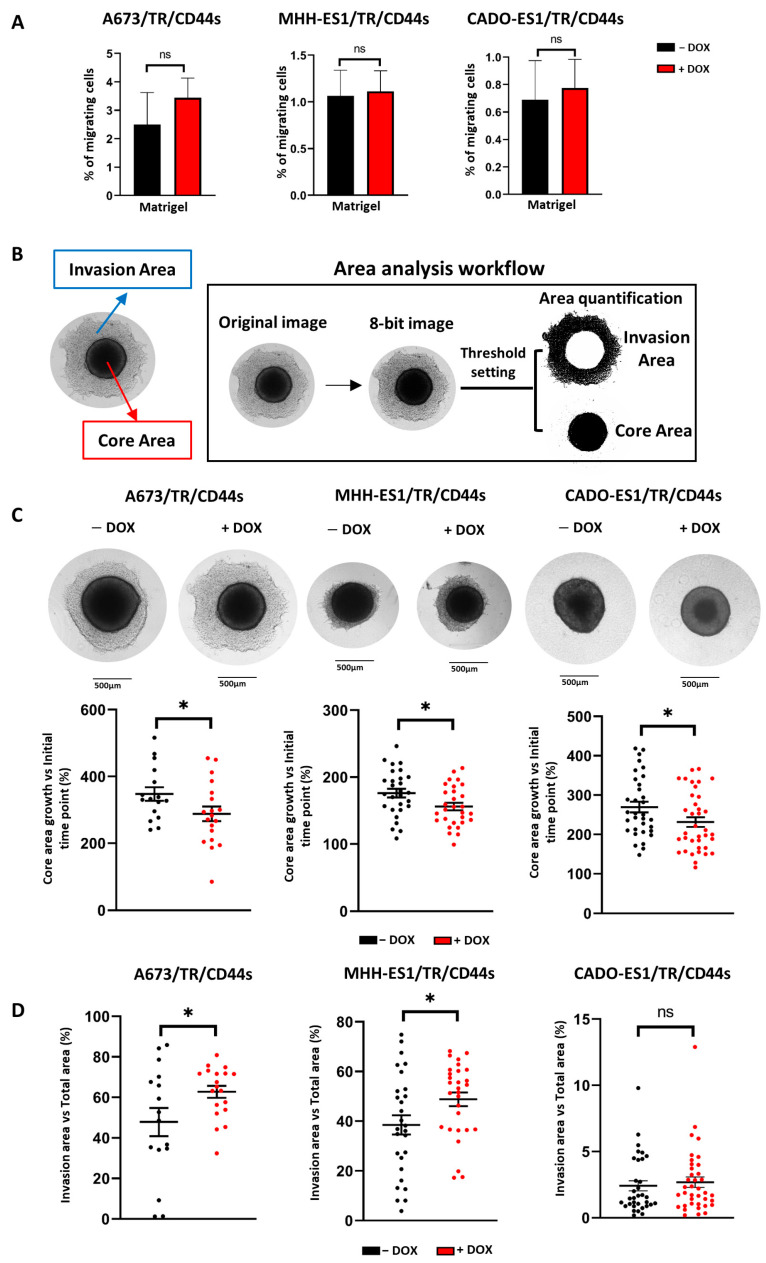
CD44s expression increases cell invasiveness in Ewing sarcoma cells. (**A**) Cell invasion assay. Ewing sarcoma cell lines were cultured in the absence or presence of doxycycline to induce CD44s expression, seeded on semi-porous membranes pre-coated with Matrigel, and the percentage of migrating cells quantified after 24 h (mean ± SD, three experiments performed in triplicate; ns: not significant). Invasion ability was not increased in any of the three Ewing sarcoma cell lines, despite CD44s overexpression. (**B**) Representative illustration of the analysis workflow used to quantify the “core” and “invasion area” of tumor spheroids. See the text for a detailed description. (**C**) Representative images of spheroids of the three Ewing sarcoma cell lines cultured in the absence or presence of doxycycline. Graphs represent the growth of the “core” area at 144 h, calculated as a ratio of the initial “core” area. Each point represents an individual tumor spheroid. (**D**) Quantification of the invasion area. The invasion area was quantified as shown in (**B**) and represented as a percentage of the total area of the spheroid (mean ± SE, ns: not significant, * *p* < 0.05; Student’s *t*-test).

## Data Availability

The datasets generated during the current study are available from the corresponding author on reasonable request. RNAseq data generated in this study have been deposited in NCBI’s Gene Expression Omnibus and are accessible through GEO Series accession number GSE232739 (https://www.ncbi.nlm.nih.gov/geo/query/acc.cgi?acc=GSE232739, updated 22 May 2023). Custom scripts and command line have been deposited in https://github.com/BU-ISCIII (22 May 2023).

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
