# Peer review of "CD44 Modulates Cell Migration and Invasion in Ewing Sarcoma Cells"

_ijms, 2023, doi:10.3390/ijms241411774_

Round 1
Reviewer 1 Report
This manuscript presented by Fernández-Tabanera et al. describes a potential role of CD44 on migration and invasion in Ewing sarcoma. They identified CD44 as a prominent EWSR1::FLI1-downregulated gene by comparing high and low EWSR1::FLI1 expression condition using doxycycline inducible EWSR1::FLI1 knockdown system. They found that among CD44 isoforms the standard CD44 (CD44s) was particularly upregulated upon EWSR1::FLI1 knockdown. Furthermore, using doxycycline inducible CD44s Ewing sarcoma cell lines they demonstrated that CD44s could contribute to migrative and invasive phenotypes, potentially by interacting differentially with low or high molecular weight hyaluronic acid.
Although this manuscript investigates scientifically important aspects in Ewing sarcoma biology and to a large part well-written, there are some concerns which should be addressed before publication.
Specific comments
Major concerns
1. Line numbers 79-82
The authors state ‘To our knowledge, the role of CD44 in Ewing sarcoma has not yet been elucidated. In consequence, it would be interesting to determine the function of CD44 in Ewing sarcoma, with a focus on their possible role in cell migration and invasiveness’. However, in the article ‘CD44 enhances tumor aggressiveness by promoting tumor cell plasticity’ published in 2015 Paulis et al. has already investigated the role of CD44 in the context of Ewing sarcoma. The authors should take the findings described in the article into consideration and discuss accordingly.
2. Line numbers 235-237
The authors state ‘…CD44s upregulation reduced the total number of cells in each well, as demonstrated by the significant reduction (range between 40–62%) in the amount of dye released after cell de-staining (p-value < 0.001, Student’s T test) (Figure 4B)’. In the opinion of this Reviewer the amount of dye release will correlate with both cell number and colony size. Accordingly, it may not be adequate to conclude ‘CD44s upregulation reduced the total number of cells in each well’, which also affects the conclusion ‘CD44s is involved in cell proliferation without modifying the clonogenic capability of Ewing sarcoma cell lines.’
Minor concerns
1. To this Reviewer it is interesting to see if the differential role of CD44 in EWSR1::FLI1 positive and EWSR1::ERG positive cell lines will be dependent on cell line (CADO-ES1) or indeed on EWSR1::ERG by using an additional EWSR1-ERG positive cell line.
2. In CADO-ES1 one can observe a heavy glycosylation of CD44 in a higher and lower molecular weight range based on the WB image, while in A673 and MHH-ES1 not. To this Reviewer it is interesting to see if the glycosylation levels will differentially affect CD44 function in Ewing sarcoma.
3. In order to corroborate the findings of CD44 function on invasion and migration by interacting with hyaluronic acid, it is interesting to see if a blocking anti-CD44 antibody can inhibit invasive and migrative phenotype of Ewing sarcoma cell lines.
Author Response
Please, see the attachment.

Reviewer 2 Report
General comment
The experimental pathway of the study is well planned and the results on the correlation of CD44 upregulation with cell migration and invasiveness in the EWS in vitro model appoint CD44 as marker of the EWS1::FLI1 low phenotype. However it remains to clarify CD44 role on EWS progression respect/with other dysregulated genes to select CD44 as potential target for treatment.
Specific criticisms
Introduction (page2 lines 57-62): the paragraph should be reworded as the sequence of the milestone discoveries is mixed up. After citing Franzetti, 2017 (Ref 4) and Aynaud, 2020 (ref 5) on the composite landscape of Ewing Sarcoma the Authors say that in “recent” years they looked for genes positively or negatively expressed in EWS cell models (ref 6-11): ref 7 and 8 are dated 2006 and 2007, respectively!
Page 2, line 81: their possible role should be “its” possible role
RESULTS Page 4 line 147 : "additionally" can be deleted
Page 4 lines 130-145 this paragraph should be summarized to be appropriate to a Results section.
Discussion Page 14 368-385 There are concepts already expressed in the Introduction which might be slimmed. Based on the obtained findings could the Authors discuss the coherence of EWSR1::FLI1low expression program, upregulation of CD44 and the mechanisms of epithelial-mesenchymal transition, extracellular matrix interaction and degradation?
Though CD44 has been catalogued as pro-metastatic marker in several malignancies, this is the first evidence in EWS, accounting for the Authors’ statement that CD44 is a potential driver of cell migration and invasion in this tumor. However CD44 intersection with the overall EWS transcription profiling is completely unknown. To this regard a comment on other genes upregulated upon EWSR1::FLI1 knockdown would be appropriate: but with the exception of CCK no other gene is mentioned as potential player and target for treatment action .
Conclusions The results obtained in this study point to the relevance of CD44 in the EWSR1::FL1 knockdown and predict CD44 might concur with other genetic and epigenetic determinants to the progression of EWS. The Authors should underline that this information is acquired using an in vitro model that simplifies the in vivo composite and dynamic situation of the coexistence of two EWS cell populations. This is one limitation of the study. In addition, even if a new challenging plug is disclosed to understand the biology of EWS, the mechanistic trajectory remains to be delineated to raise translational opportunities.
minor editing
Author Response
Please, see the attachment
